# Interactive Viewpoint Suggestion with Deep Classification Network for Browsing 3D Object Galleries

Author's affiliation

## ABSTRACT

Viewpoint selection techniques have been one of the most fundamental and main topics in computer graphics for many years and closely related to some other applications, such as data visualization or camera placement. Also, according to the rapid improvements in the deep learning field, some viewpoint selection techniques based on deep learning methods have been proposed. However, these methods are typically only focused on suggesting one viewpoint of a single object. In this paper, we propose a new framework for simultaneously selecting viewpoints to compare multiple objects in 3D galleries. Our network takes rendered images of each object from various viewpoints as inputs, and outputs optimal viewpoints for each object so that users can easily grasp characteristics of 3D objects. Furthermore, for more general-purpose usage, the system also supports interactions with users, so that users can easily explore the viewpoints of some objects. We validated the efficiency of our approach through the user study.

**Index Terms:** Human-centered computing—Visualization—Visualization systems and tools—Visualization toolkits; Human-centered computing—Visualization—Visualization design and evaluation methods

## 1 INTRODUCTION

3D object galleries (e.g., Sketchfab[1], Free3D[2] and ShapeNet dataset [5]) have been released all over the world, increasing the opportunities for us to make 3D artwork. Such galleries show a series of images rendered from fixed viewpoints (e.g., frontal view), but it may not show important parts of each 3D model (e.g., back view) since 3D objects are generally given in arbitrary scale, position and orientation in 3D-space. In such case, users must load 3D models into 3D software and check them one-by-one while changing viewpoints, which is very time-consuming.

Against this background, viewpoint setting is one of the basic operations in 3D, but many researches to automatically (or semi-automatically) set viewpoints using human perception [6, 15] and shape characteristics [9, 18] have been proposed. Especially, with the development of deep learning techniques, convolutional neural networks (CNNs)-based approaches are attracting attention since CNNs can automatically compute appropriate features from each viewpoint, instead of heuristics used by human designers [14, 20]. However, the goal of the above existing methods is to independently find one viewpoint for an input "single" object. That is, even if a good viewpoint can be selected for an individual object, in a situation where multiple objects in 3D galleries are compared (e.g., object A vs object B, ..., vs object Z), the suggested viewpoints are not suitable for representing the differences between objects.

To address this problem, we propose a deep learning-based

---

*e-mail: Author's e-mail

[1] https://sketchfab.com/
[2] https://free3d.com/3d-models/

method for interactively suggesting suitable viewpoints for comparing multiple objects in 3D galleries. We assumed that the recognizability of each object and the similarity of viewpoints between objects are important. For optimizing the set of viewpoints, we use pre-defined viewpoints to render the 3D objects. Then, each viewpoint of each object are represented as a feature vector using a deep learning network. Using these feature vectors, we optimize and find a good combination of viewpoints that maximizes a better balance between recognizability of each object (= easy to identify) and similarity of viewpoints between objects. (= easy to compare)

We conducted user studies to validate the effectiveness of the proposed method. In the user studies, we gave participants a task to find objects from our prepared 3D galleries using the suggested viewpoints by the proposed system, and which is compared with the baseline methods. In addition, we gathered users' feedback on the usability of the proposed method.

The contribution of our work is summarized as follows:

1. A concept to interactively suggest viewpoints suitable for comparing 3D objects in 3D galleries.

2. A method to suggest viewpoints with a better balance between the recognizability of each object and the similarity of objects' viewpoints, based on CNN-based features.

3. User studies to evaluate the effectiveness and the usability of our method from qualitative perspectives.

## 2 RELATED WORK

This section reviews prior work on optimizing viewpoints using (1) heuristic approaches and (2) deep learning approaches.

### 2.1 Heuristic Approach

To find preferable viewpoints for 3D objects, one approach is to use shape characteristics [10]. For example, Vázquez et al. [23] regarded the visibility of the input object as a probability in the ordinary information theory, and defined characteristics of each viewpoint. In addition, Lee et al. [16] compute a measure of regional importance for 3D objects and propose a saliency-guided view selection that maximizes the sum of the saliency for visible regions of the object. Although their systems enable users to change features obtained from each view, it remains difficult to decide which features are appropriate to use. Therefore, some researches to investigate the relationship between the viewpoint qualities and features extracted from render images have been proposed [3, 4]. Secord et al. [19] prepared existing heuristics (e.g., curvature) and constructed a simple regression model to find one viewpoint based on a user survey. While these are suitable for capturing the most salient attributes of the single object, they are less suited to compare the attributes of multiple objects in 3D model galleries.

### 2.2 Deep learning-based Approach

Applying deep learning techniques to the viewpoint selection task allows users to skip the process of preparing features in advance. We therefore consider a deep learning model for extracting 3D object characteristics [24] here. Su et al. [21] presented a method to compute a view-saliency from multi-view images, and viewpoints

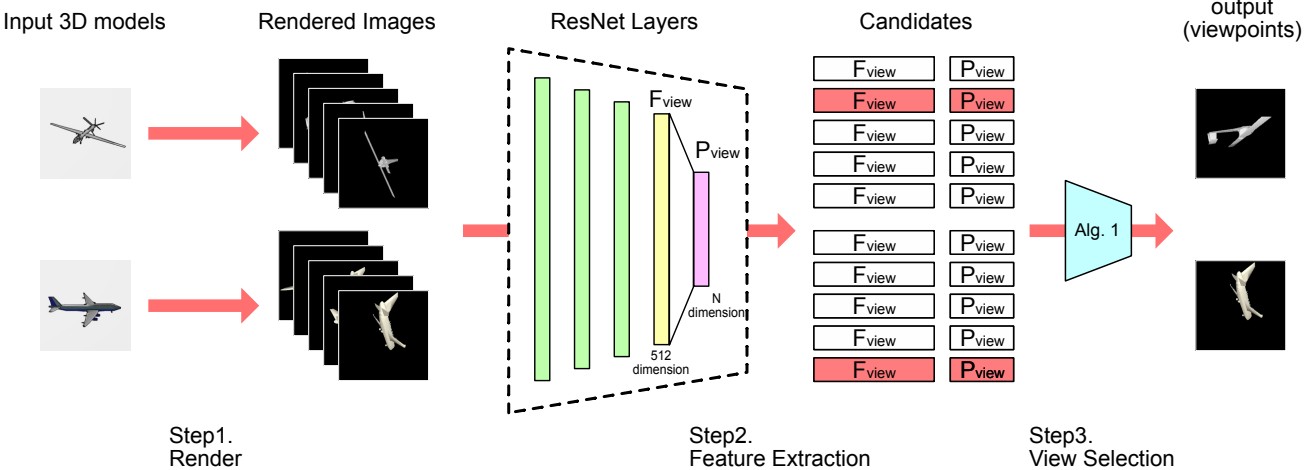

Figure 1: System outline. Given 3D object galleries, the system first (1) renders $i$-th object from $j$-th view candidates (648 views) and then (2) computes a feature vector $\boldsymbol{F}_{view}$ and a vector of label probabilities $\boldsymbol{P}_{view}$ for each viewpoint based on the ResNet-18 model [11]. Based on the obtained vectors, viewpoints of all objects are selected from the view candidates automatically and simultaneously.

with high view-saliency contain many features related to object classes. That is, their goal is to estimate object classes tags (e.g., airplane, car, and ship), but it is highly relevant to the viewpoint selection problem. Song et al. [20] focused on the consistency of characteristics between an input 3D object and its images rendered from multiple views, and trained a CNN model which estimates 3D saliency maps on the 3D object under an unsupervised manner, named UMVCNN. Based on the estimated saliency maps, they also find one viewpoint which maximizes the sum of the saliency for visible regions, as with existing viewpoint selections [10, 16, 22].

A common problem with existing methods is to individually suggest one viewpoint of the input "single" object. The suggested viewpoint is unsuitable for comparing many 3D objects on 3D model galleries since they did not consider relationships between 3D objects (e.g., airplane A vs airplane B). In addition, existing systems build models which classifies 2D images into object classes in advance, but modifying the estimation results require re-training the model, making it inappropriate for user editing.

Then, we build their system but use it to extract features from the rendered images. The proposed method has the advantage of interactive editing without a relearning process.

## 3 METHOD

Fig. 1 shows the outline of our viewpoint selection system. A user first inputs multiple 3D objects, and the system simultaneously optimizes their viewpoints. By observing rendered images of the input 3D objects from the suggested viewpoints, the user can easily and efficiently browse and compare between 3D objects. However, the main objective is not to fully-automatically select all viewpoints of 3D objects, but rather to provide a good starting point for manual exploring. For this, we additionally develop a function to edit initial viewpoint results and re-optimize all viewpoints with user-specified constraints. The user can repeat this process until each viewpoint is complete and acceptable. Note that we are assuming that no prior alignment is taken between objects.

### 3.1 Scene Setting

As with existing viewpoint selection systems [13, 20, 22], we sample 162 virtual cameras uniformly distributed on a 3D viewing sphere. Next, we rendered four attitudes rotated at $90°$, $180°$, and $270°$

around the direction of each camera and generate 648 view candidates $j \in \{0, \cdots, M-1\}$. The direction of each camera is a direction vector from the camera position to the origin of the input models. As a rendering setup, we assigned a fixed material (= cream color) to the input models, a black color to the background, and the light sources were placed outside of each camera. The reason of this color setting is if we have different color settings among given objects it could be a shortcut for the model to use that color information for prediction, and view suggestion system might not work properly.

### 3.2 Model Architecture

Our aim is to distinguish which object each rendered image is. Then, we focused on the ResNet-18 model [11], which is well-known in the image classification task. First, we assigned label IDs (e.g., airplane A and airplane B) to rendered images of the input 3D objects, and build a model to estimate the label IDs from the rendered images. To improve the efficiency of the training and avoid overfitting, we employ a transfer learning from a pre-trained ResNet-18 model with ImageNet dataset [8]. All weights in ResNet-18 model are fixed, and we added the last layer to predict each object's label ID. In addition, we extract a feature in each view for each object, named view features $\boldsymbol{F}_c$, from the second last layer of the trained model. Note that $\boldsymbol{F}_c$ can be represented as a 512-dimensional vector.

We describe the detailed settings for training. We used a GPU on Google Colaboratory to retrain the model. The batch size was set to 64. Since all training converged well after 10 epochs, we used the model after 10 epochs of training for all the experiments in this study.

### 3.3 View Scoring

Our goal is to suggest a viewpoint of each object $c_i \in \{0, \cdots, N-1\}$ for browsing multiple 3D objects in input galleries, so it is necessary to consider viewpoints that (1) make it easy to see the objects' characteristics and (2) make all viewpoints roughly similar.

Then, we define the following energy function based on the recognizability of each object (easy-to-identify) and while preserving the viewpoint similarity between objects (easy-to-identify).

$$E_{total} = \lambda_{recog}E_{recog} + \lambda_{alike}E_{alike} \qquad (1)$$

where $\lambda_{recog}$ and $\lambda_{alike}$ are weight values. In this paper, we empirically found $\lambda_{recog} = -450.0$ and $\lambda_{alike} = 1.0$ provided a good balance. Note that the reason why $\lambda_{recog}$ is a minus value is that larger values of $E_{recog}$ mean better.

The $E_{recog}$ is a term that represents the average of the correct classification rate for each object's rendered image, defined as follows:

$$E_{recog} = \frac{1}{N} \sum_{i \in [N]} prob_{c_i} \qquad (2)$$

where $c_i$ is the viewpoint selected for the $i$-th object and $N$ is the amount of 3D objects in the input gallery. The $prob_{c_i}$ means how well the viewpoint $c_i$ represents the features of the $i$-th object. However, the classification rate is not smooth on the viewing sphere, which makes many local minimum. Then, we simply smooth the score of classification rate on the view sphere, given by

$$prob_{c_i} = \min(S_{cap}, Score_{c_i}) \qquad (3)$$

$$Score_{c_i} = \frac{\sum_{c \in [M]} g(d_q(c_i, c)) \cdot \mathrm{softmax}(P_i(c_i))}{\sum_{c \in [M]} g(d_q(c_i, c))} \qquad (4)$$

where $S_{cap}$ is the constant value used for clipping the $prob_{c_i}$ (In this paper, $S_{cap} = 0.7$), $M$ is the amount of view candidates ($= 648$ views), and the $d_q$ is the quaternion distance between two viewpoints ($=$ camera angles). The $g(\cdot)$ represents the Gaussian function (In this paper, we empirically set its variance $\sigma^2$ to 20.0) and the $\mathrm{softmax}(\cdot)$ is the softmax function. $P(c)$ is a $N$-dimensional vector computed from a viewpoint $c$, and consists of the probability value $\mathrm{softmax}(P_i(c)) \in [0.0, 1.0]$ that our model classifies a 2D rendered image as the $i$-th object. That is, when this value is high, it means that the recognizability at the viewpoint $c$ is high. Note that the idea of using the classifier's estimation results was inspired by Su et al. [21]. Please refer to it for more details.

Although $E_{recog}$ can find appropriate viewpoints for estimating the label IDs from rendered images of 3D objects, each viewpoint ($=$ camera angle) can be very different. Therefore, to make the viewpoint of each object similar ($=$ easy to compare), we define a similarity of the viewpoint features $E_{alike}$ as follows:

$$E_{alike} = \frac{1}{N(N-1)} \sum_{i \in N} \sum_{j \neq i} d_E^2(\boldsymbol{F}_{c_i}, \boldsymbol{F}_{c_j}) \qquad (5)$$

where $d_E(\boldsymbol{F}_{c_i}, \boldsymbol{F}_{c_j})$ means the Euclidean distance between two view features obtained from the viewpoint of $i$-th object and the viewpoint of $j$-th object.

### 3.4 Viewpoint Optimization

A exhaustive search is the simplest method to find $\{c_i\}$ that minimizes $E_{total}$, but takes too long to find viewpoints for 3D object galleries: its time complexity is $O(M^N)$. The main reason is that $E_{alike}$ term depends on the relationship with all viewpoints, which is difficult to optimize directly. Then, we consider an approximate approach to minimize the sum of distances between each view feature and the average value of features by iteratively performing the random initialization and view re-assignment, referring to the $k$-means method [17]. In addition, to consider the $E_{recog}$ term, the $prob_{c_i}$ is included in the re-assignment step, and we adjusted coefficients to be consistent with our initial definition of $E_{total}$. Algorithm 1 shows a pseudo-code of the approximate method. Note that we run the above iterative optimization process multiple times and the best result were used. In this paper, we empirically set the number of iterations, $A_1$ and $A_2$, to 20 and 4 respectively.

In addition, we have developed a function that allows users to manually set viewpoints for some objects. The user first specifies some objects, and the system then re-performs the optimization process again with the user-specified viewpoints as constraints. Based on this function, the system can better reflect user preferences.

**Algorithm 1** Optimize $E_{total}$

---

$E_{result} \leftarrow \mathrm{Float.Max}$
$\{R_i\} \leftarrow \{S_i\} \in \mathbb{R}^N$
**for** $\mathrm{ite}_1 \leftarrow 1$ to $A_1$ **do**
  $\{S_i\} \leftarrow \texttt{random Initialize}$
  **for** $\mathrm{ite}_2 \leftarrow 1$ to $A_2$ **do**
    $Ave \leftarrow \texttt{Average of } \{\boldsymbol{F}_{s_i}\}$
    **for** $i \leftarrow 1$ to $N$ **do**
      **if** $S_i$ is fixed by user **then**
        $S_i \leftarrow \texttt{user's selection}$
      **else**
        $S_i \leftarrow \arg\min_c\{\lambda_{recog} prob_c + N\lambda_{alike} d_E^2(Ave, \boldsymbol{F}_c)\}$
      **end if**
    **end for**
  **end for**
  Calculate $E_{total}$ with current $\{S_i\}$
  **if** $E_{total} < E_{result}$ **then**
    $\{R_i\} \leftarrow \{S_i\}$
  **end if**
**end for**
**return** $\{R_i\}$

---

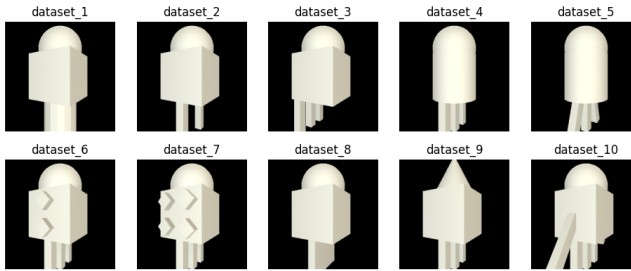

Figure 2: Primitive dataset created manually by authors using Open SCAD.

## 4 USER STUDY

We conducted a user study to evaluate the effectiveness and the usability of our system.

### 4.1 Datasets

We first prepared two datasets of 3D objects; (i) airplane and (ii) primitive. In case of the airplane dataset, we extracted 10 objects tagged "airplane" from the ShapeNet dataset [5]. However, a fully-random selection might obtain almost indistinguishable objects, so we selected to avoid such combinations. In addition, as mentioned in Sect. 3.1, we removed the original material of each object and assigned a fixed color (cream color) to all airplane objects. For primitive dataset, we make 10 simple artificial objects by combining multiple primitives (e.g., cylinders and cubes) using Open SCAD[3]. All objects in the primitive dataset have a similar structure, but each object has unique features which appear at different viewpoints (see Fig. 2). Note that each object in the datasets was randomly rotated to generate unaligned 3D galleries.

### 4.2 System Setup

Fig. 3 shows the screenshot of our implementation. The system screen is divided into three main areas: (i) the instruction area, (ii) the object area, and (iii) the tool button area.

---

[3] https://openscad.org/

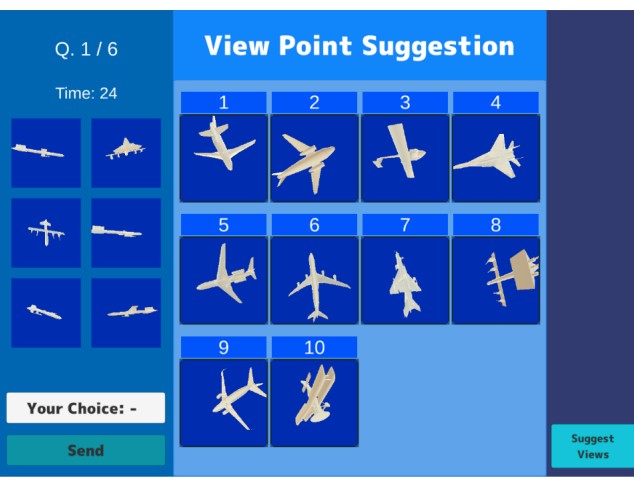

Figure 3: Screenshot of our interface used in the user studies. (Left) the instruction area, (Middle) the object area, and (Right) the tool button area.

The instruction area shows the operation time, the question IDs, and instructions for each question. We also placed a "send" button at the bottom for submitting participants' answers. The object area shows target 3D objects that users compare and at the start of use, the viewpoint of each object is assigned randomly. By dragging on the screen displaying each object, user can freely move its corresponding camera. And, by manipulating the viewpoint while holding down the control key, users can fix their viewpoints of objects. After users fixed viewpoints, the candidate viewpoint closest to the camera position at that time is used as user's selection. In the tool button area, we placed the viewpoint selection button. When pressing the button, the system assigns the suggested viewpoints into all viewpoints of 3D objects visualized in the object area. Note the computation time of all methods can be completed in less than a few seconds but we add a progress bar to visualize the progress of the computation for reducing users' psychological stress.

For comparison, we prepared three baseline methods: (i) RECOG ONLY, (ii) ALIKE ONLY, and (iii) RANDOM. The RECOG ONLY and the ALIKE ONLY select viewpoints based on the first term and the second term in Equation 1, respectively. The RANDOM is to randomly select viewpoints from view candidates.

### 4.3 Tasks

The user study was divided into two parts, named a QUIZ task and a FREE task. In the QUIZ task, we first provided the participants with 6 images of one object rendered from random viewpoints. Next, the participants were asked to find one object represented the 6 images from 3D object galleries (= airplane/primitive datasets). The purpose of the QUIZ task is to verify how useful the suggested viewpoints are in comparing 3D objects. After using each method, the participants answered a question: "*whether the suggested viewpoints with each method are suitable for comparing 3D objects*" using a five-point Likert scale (from 1: Strong disagree to 5: Strongly agree).

In the FREE task, the participants were asked to explore viewpoints which enables users to successfully browse 3D galleries. The FREE task was performed twice for our method and one of the three baseline method per participant. We assigned randomly each baseline methods to each participant to balance the number of participant. The main reason is that the goal of the FREE task is to investigate the advantages of viewpoint selection method itself unlike the QUIZ task. Then, we focused on gathering feedback from the participants rather than quantifying differences in methods. After the FREE task, the participants also answered a questionnaire about their satisfac-

tion and their impression. This questionnaire was not mandatory. The questionnaire was made based on the Google Form and the question items will be described in Sect. 4.5.

The total evaluation process took approximately 90 minutes per participant. We paid a fixed reward for our view setting task regardless of the quality, i.e., 10.00 USA. Note that we assigned the order of the four methods used in the QUIZ task and the methods in the FREE task to each participants to avoid bias. We randomly assigned the names TYPE1 - TYPE4 to the four methods during each task to conceal which method participants were actually using, to avoid the effect of bias. For datasets, we assigned the two types of datasets to each participant to balance the number of participant. Also, all participants used different datasets for the two types of tasks.

### 4.4 Participants and Apparatus

We invited 12 participants (P1, P2, $\cdots$ P12). All participants have much experience in using computers in their daily use and all except P7 have computer science background. P5, P6, P8, P11, & P12 were also familiar with manipulating 3D objects using Unity[4], Blender[5], Revit[6] and CAD[7]. The rest did not report any expertise in 3D software.

We provided the participants with the instruction documents of the user study and the link to our WebGL application deployed on the server. After receiving the instruction, the participants could manipulate our application on their own device.

### 4.5 Results

Fig. 4 shows the post-experiment questionnaire result of the QUIZ task. According to these results, the participants answered that the proposed method was better than all baseline methods in the case of the airplane dataset, but RECOG ONLY was better in the case of the PRIMITIVE dataset. From this result. depending on the type of object, it may be more effective to use only $E_{recog}$ and not $E_{alike}$. We also see that the result of ALIKE ONLY was the worst for both datasets. It is thought that since all viewpoints are similar, ALIKE ONLY might not be useful for observing important parts of each object. In addition, in case of the PRIMITIVE dataset, the results of the RANDOM method are not rated low and are comparable to the proposed method. We will now discuss the P12 comments about the reasons for this result in more detail.

- *The suggested viewpoints are changed with each press of the "suggest views" button, which may be good if we want to see each object from multiple viewpoints or if we want to change the viewpoints.*

This comment suggests that the other baseline methods (i.e., RECOG ONLY and ALIKE ONLY) tend to optimize one viewpoint of each objects and participants who wanted to see each object from multiple viewpoints were not satisfied with them.

The FREE task is mainly for gathering feedback and comments from participants, but we additionally asked the participants to answer three additional questionnaires at the end of FREE task: (1) about the quality of the suggested viewpoints based on a five-point Likert scale, (2) about the usability based on the System Usability Scale (SUS) [2], and (3) about the quality of viewpoint quality, user editing function and so on.

Fig. 5 shows these results on (1) and (2). From the result (1), we can see that the proposed method can suggest better viewpoints than the baseline methods. In addition, our system's SUS score (means values) is 75.63, which is regarded as "good" and has a grading scale of Grade B [1]. Note that the RECOG ONLY's score, the ALIKE

---

[4] https://unity.com/
[5] https://www.blender.org/
[6] https://www.autodesk.com/products/revit
[7] https://www.autodesk.com/solutions/cad-software

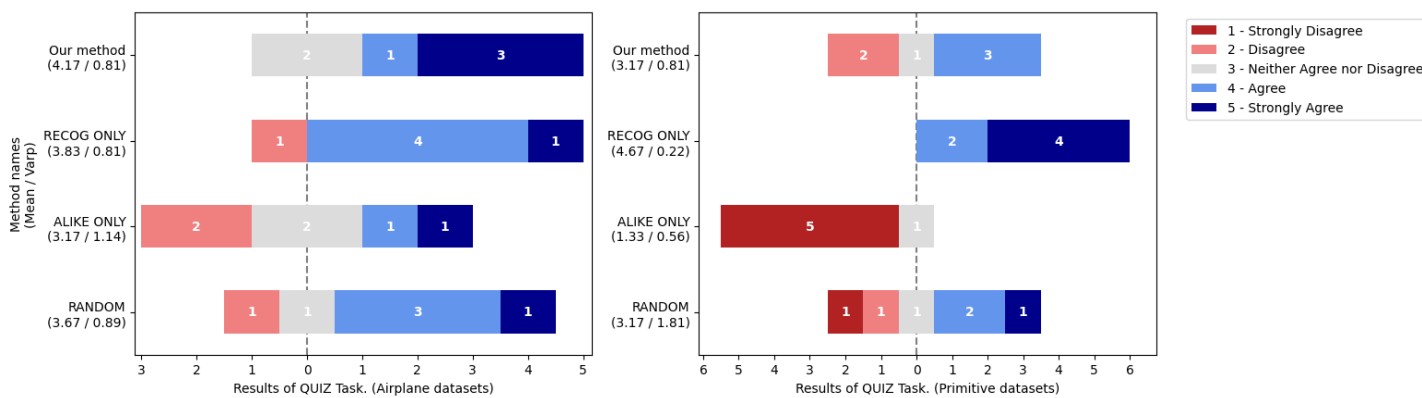

Figure 4: Results of the QUIZ Task using (Left) the Airplane dataset an (Right) the Primitive dataset.

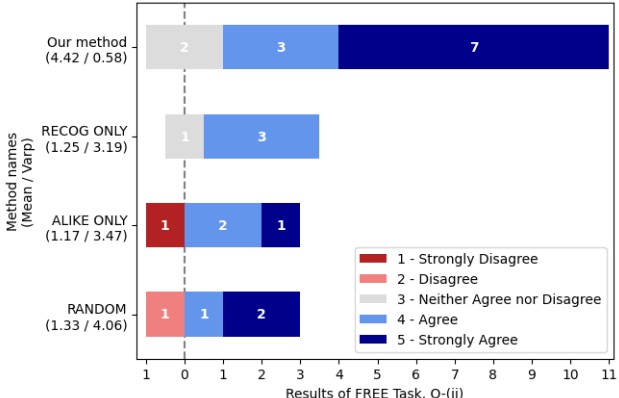 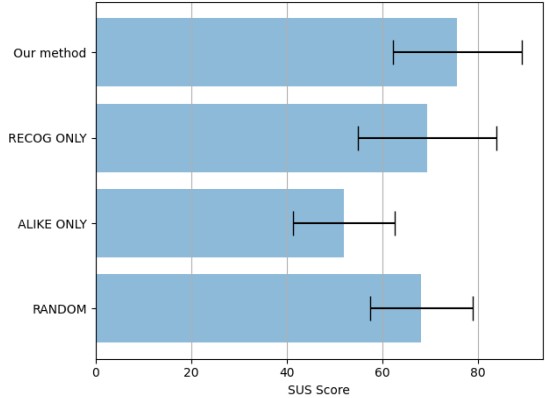

Figure 5: Results of the FREE Task using a five-point Likert scale and the System Usability Scale (SUS) [2]. Note that all 12 participants used our method, while each baseline method was evaluated by four participants.

ONLY's score, and the RANDOM's score is 69.374, 51.875 68.125, respectively.

Next, we will now discuss the comments on (3) in more detail. In case of the quality of the suggested viewpoints, about half of the participants (P1, P3, P4, P5, P7, P9, P10, P11 & P12) rated the results of the proposed method as "extremely satisfied." For example, P3 said that *"I think the suggested viewpoints were good because many 3D objects were easy to visually compare."* and P9 commented that *"The suggested viewpoints captured many of the characteristics and I felt that manual rotation was not really necessary."* In addition, two participants mentioned that *"The first suggestion was easy to compare visually because it was from a top viewpoint"* (by P3) and *"It was easy to understand, especially since it was a low angle where you could see the engine part of the plane"* (by P10). That is, The proposed method allows us to suggest viewpoints similar to each other by adding $E_{alike}$. In contrast, P2 and P6 responded that although the suggested viewpoint (ours without user-modification) made it difficult to compare some objects and had to be modified, its results are both sufficient as a starting points. However, since the proposed method and the baseline methods does not directly take semantic features into account, the suggested viewpoints may be unnatural. In the user study, such comments were received such as *"Some airplanes are upside down, which is not a common pose"* (by P8). In the future, we will extend our method to include them.

Regarding the user editing function, several participants indicated that it was very effective in optimizing all the remaining viewpoints based on the user-specified few viewpoints (constraints). For exam-

ple, *"After setting three viewpoints manually, the system automatically suggests the other viewpoints aligned with it. It is very useful"* (by P4 and P6). On the other hand, some participants did not find the user editing function useful. P10 commented, *"I thought that when I set up a view that captured a feature of one object (e.g., the engines on the right side of the airplane), I would expect other object views to capture the same feature. But no such viewpoint were suggested perfectly, so it is a little confusing."* This may be related to semantic features as described above.

Lastly, there were some requests from participants to add functions, as follows:

- *"It seems difficult (even for humans) to automatically suggest optimal viewpoints that enable us to visually compare many 3D objects at once, so I thought it might be better if we can focus on a few objects: we can select two objects, and the system suggests viewpoints considering only those two objects and make it easier to compare those two objects clearly"* (by P10).

- *"It would be better to have a function that allows the system to present multiple viewpoints and choose one of them [12]"* (by P11).

According to these comments, the participants also identified several issues with the current system, but we found those not to be serious problems and it is possible to easily solve them with engineering effort.

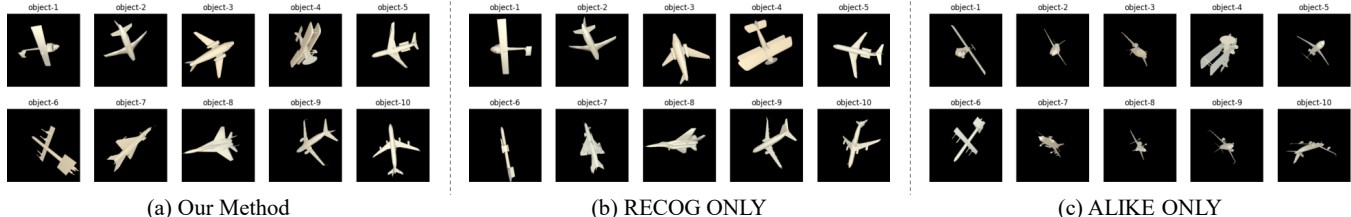

| (a) Our Method | (b) RECOG ONLY | (c) ALIKE ONLY |

Figure 6: An example of the suggested viewpoints for the airplane datasets. (a) our method, (b) RECOG ONLY, and (c) ALIKE ONLY.

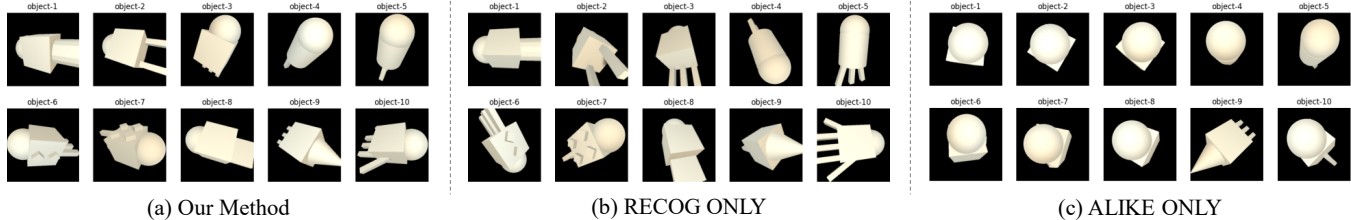

| (a) Our Method | (b) RECOG ONLY | (c) ALIKE ONLY |

Figure 7: An example of the suggested viewpoints for the primitive datasets. (a) our method, (b) RECOG ONLY, and (c) ALIKE ONLY.

## 5 EXAMPLE OUTPUTS

We compare the viewpoints suggested by the proposed method with the baseline methods (described in Sect. 4) as an ablation study. Fig. 6 and Fig. 7 shows examples of rendered 3D objects from the suggested viewpoints.

From these results, ALIKE ONLY enables users to see 3D objects from roughly-similar camera angles, but this is difficult for us to visually compare 3D objects with each other. In case of RECOG ONLY, it is possible to find optimal viewpoints that captures shape characteristics of the objects, but each viewpoint is different from each other. In contrast, our method also enables users to make a good balance between shape characteristics capturing and similarity of each viewpoint. For example, in case of the airplane datasets, our method suggests viewpoints of the Object-6 and the Object-8 from the top views, and the results of RECOG ONLY are from side views. However, we can see it remains difficult to visually compare between the Object-4 and the Object-5 in the primitive dataset.

## 6 FUTURE WORK

Our issue is that the suggested viewpoints were not "perfectly" aligned. For example, the airplane dataset were rendered from top views and the primitive dataset were rendered from side views, but their camera angle were slightly different. The main reason is that the prepared pre-trained ResNet (described in Sect. 3.2) is robust to rotation and the feature vectors $F_c$ might not include differences in rotation information around the direction of each camera. We plan to combine such rotational information with our method in the future.

The present paper focuses mainly on comparing static objects, but it may be interesting to explore the possibility of extending it to animation data browsing [7].

## 7 CONCLUSION

This paper has presented a deep learning-based method for interactively suggesting suitable viewpoints for browsing multiple objects in 3D galleries. We focused on the recognizability of each object in 3D galleries (= easy-to-identify) and the similarity of viewpoints between objects (= easy-to-compare), and proposed a method with features used for 3D object classification in the field of deep learning, instead of shape characteristics. Our user study showed that

the proposed method outperforms the baseline method for some datasets.

## ACKNOWLEDGMENTS

We thank the anonymous reviewers and the editor for their insightful comments that improved this manuscript. Other acknowledgements removed for anonymous review.

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
