# OpenReview forum: "Interactive Viewpoint Suggestion with Deep Classification Network for Browsing 3D Object Galleries"
_graphicsinterface.org/Graphics_Interface/2023/Conference — Submitted to GI 2023_

### Official Review · Reviewer_TUts · 2023-01-01
**low validity and novelty**

**Rating:** 4
**Confidence:** 4

**Review:**

This paper presents a deep neural network based framework that suggests viewpoints for browsing 3D objects. The topic is important and falls in the interests of the GI community. The paper is well motivated to reduce human labor in determining the best set of viewpoints for 3D objects. The related work for this paper is short, covering two different types of approaches for optimizing viewpoints for 3D objects. I’d like to see more relevant HCI papers given the submitted venue.

In general, the presentation of the proposed algorithm is clear. However, it is unclear why two rationales are considered in designing the algorithm: “(1) make it easy to see the objects’ characteristics and (2) make all viewpoints roughly similar.” I especially had a hard time to understand why we want to make all viewpoints roughly similar. Also, why these two considerations? What are other considerations that might be missed? This rationale part needs to be strengthened. The novelty of the technique is not very high. It is based on well-known existing networks. Perhaps the most novel part is the view scoring, which needs better rationales and emphasis.

A user study is conducted to assess the effectiveness of the proposed technique. The results show some marginal improvements compared to a Random baseline and some variations of the algorithm. The user study design has serious issues, and in other words, the validity of the paper is low. First, the random baseline is too low given that there exist several algorithms in the literature. The study should choose a stronger baseline. Second, only 6 images are used in each of the QUIZ and FREE tasks. It is simply too few! And there is only one subjective measure:  whether the suggested viewpoints with each method are suitable for comparing 3D objects.” Only 6 data points for comparing the four techniques in each task (Fig 4). It also doesn’t make sense to use the chart like Fig 5-left to compare the techniques because their total numbers of responses are different. Third, the results do not show any notable advantages for the full-version of the technique compared to the random baseline, including both the questionnaire and SUS scores. The paper should explain this and focuses more on the qualitative results, if the quantitative results can’t show notable benefits. Lastly, I think this paper will be benefited from a large scale crowdsourcing study (e.g., using MTurk) with many more participants and images/tasks.

In sum, I think this paper is not ready for publication in its current form.

---

### Official Review · Reviewer_K8db · 2023-01-13
**Nice results, timid evaluation**

**Rating:** 6
**Confidence:** 1

**Review:**

The article introduces a novel technique to generate several viewpoints from a collection of  3d models. The main goal is to select viewpoints that allow distinguishing one particular shape from the dataset to another one.

I am not a learning expert, so judging and reviewing the main technical part is hard. Nevertheless, I like the validation (while I would like to see a larger population sample and a more diverse pool). I think the results are convincing, and it seems that the proposed technique helps discriminate between shapes.

---

### Official Review · Reviewer_Ac8F · 2023-01-14
**The paper describes a method to find and suggest viewpoints of 3D images that find a balance between recognizability of each object and effective comparison with similar objects. The proposed system also provides an interactive interface to allow manual exploration. The method was evaluated using a design study with 12 participants.**

**Rating:** 5
**Confidence:** 3

**Review:**

The authors describe a technique to find viewpoints for 3D images that offer a balance between the ability to recognize 3D images and also to effectively compare them with similar 3D objects. The authors do not do a good job of motivating the need for this approach. The introduction mentions that the authors “assumed that the recognizability of each object and similarity of viewpoints between objects are important”. It is unclear why the assumption is made. The authors do not base their approach on expert feedback or from any results obtained from focus groups, design studies etc. In addition, the choice of the airplane and primitive datasets for the user study, also seems to be arbitrary. Overall, this is an interesting work that needs more refinement and review which may not be possible in the given time.

Minor issues: There are several grammatical issues such as “many researches to automatically…” (Introduction), “A exhaustive search…” (Section 3.4) and several more.

---

### Meta-Review · Area_Chair_iedg · 2023-01-14

**Recommendation:** 4
**Confidence:** 4

**Metareview:**

In summary, this paper received 2 negative ratings and 1 positive rating: i.e., "Marginally below acceptance threshold", "Marginally above acceptance threshold", and "Ok but not good enough - rejection".


Strengths:
+ interesting topic
+ the presentation of the proposed algorithm is generally clear
+ results seem convincing

Weaknesses:
- unclear assumption of: "the recognizability of each object and similarity of viewpoints between objects are important"
- the choice of study dataset seems arbitrary
- marginal improvements compared to a Random baseline and some variations of the algorithm
- too few data points collected in the study to generalize the conclusion
- lack of HCI literature; relatively short related work

Overall, while interesting, this paper seems not ready for publication within the review cycle, because additional user study and analysis are needed. I recommend author consider the feedback in the reviews and resubmit this paper to a future venue.